# Lung and Intercostal Upper Abdomen Ultrasonography for Staging Patients with Ovarian Cancer: A Method Description and Feasibility Study

**DOI:** 10.3390/diagnostics10020085

**Published:** 2020-02-05

**Authors:** Maciej Stukan, Antonio Bugalho, Amanika Kumar, Julita Kowalewska, Dariusz Świetlik, Natalia Buda, Małgorzata Pietrzak-Stukan, Mirosław Dudziak

**Affiliations:** 1Department of Gynecologic Oncology, Gdynia Oncology Center, Pomeranian Hospitals, 81-519 Gdynia, Poland; 2CUF Infante Santo Hospital, CUF Descobertas Hospital, NOVA Medical School, 1350-070 Lisbon, Portugal; 3Department of Obstetrics and Gynecology, Division of Gynecologic Surgery, Mayo Clinic, Rochester, MN 55902, USA; 4Department of Radiology, Pomeranian Hospitals, 81-519 Gdynia, Poland; 5Department of IT and Biostatistic, Gdańsk Medical University, 80-211 Gdańsk, Poland; 6Internal Medicine, Connective Tissue Diseases and Geriatrics Department, Medical University of Gdańsk, 80-210 Gdańsk, Poland; natabud@wp.pl; 7Department of Obstetrics and Gynecology, Medicover Medical Center Gdansk, 80-309 Gdańsk, Poland

**Keywords:** lung ultrasonography, pleural cavity, diaphragm, ovarian cancer, diagnostic imaging

## Abstract

A detailed transabdominal and transvaginal ultrasound examination, performed by an expert examiner, could render a similar diagnostic performance to computed tomography for assessing pelvic/abdominal tumor spread disease in women with epithelial ovarian cancer (EOC). This study aimed to describe and assess the feasibility of lung and intercostal upper abdomen ultrasonography as pretreatment imaging of EOC metastases of supradiaphragmatic and subdiaphragmatic areas. A preoperative ultrasound examination of consecutive patients suspected of having EOC was prospectively performed using transvaginal, transabdominal, and intercostal lung and upper abdomen ultrasonography. A surgical-pathological examination was the reference standard to ultrasonography. Among 77 patients with histologically proven EOC, supradiaphragmatic disease was detected in 13 cases: pleural effusions on the right (*n *= 12) and left (*n *= 8) sides, nodular lesions on diaphragmatic pleura (*n *= 9), focal lesion in lung parenchyma (*n *= 1), and enlarged cardiophrenic lymph nodes (*n *= 1). Performance (described with area under the curve) of combined transabdominal and intercostal upper abdomen ultrasonography for subdiaphragmatic areas (*n *= 77) included the right and left diaphragm peritoneum (0.754 and 0.575 respectively), spleen hilum (0.924), hepatic hilum (0.701), and liver and spleen parenchyma (0.993 and 1.0 respectively). It was not possible to evaluate the performance of lung ultrasonography for supradiaphragmatic disease because only some patients had this region surgically explored. Preoperative lung and intercostal upper abdomen ultrasonography performed in patients with EOC can add valuable information for supradiaphragmatic and subdiaphragmatic regions. A reliable reference standard to test method performance is an area of future research. A multidisciplinary approach to ovarian cancer utilizing lung ultrasonography may assist in clinical decision-making.

## 1. Introduction

The first-line treatment for primary epithelial ovarian cancer (EOC) is debulking surgery, with the goal of removing all macroscopic disease, followed by adjuvant chemotherapy [1]. Before surgery, there are two main considerations: first, whether complete cytoreduction is possible; and second, estimation of the risk of major complications when complex surgery is planned. Presumed stage and surgical complexity, together with patient-related and disease-related variables are important to estimate severe postoperative complications after primary debulking surgery [2,3]. For advanced EOC where resection to residual disease of 1 cm or less is unlikely or the risk of complication is high, neoadjuvant chemotherapy and interval debulking surgery are associated with improved survival and reduced perioperative morbidity compared to primary debulking surgery [4].

Reliable pre-surgical predictors of resectability would be valuable tools for assigning patients to the best management plans [4]. It has been shown that whole-body diffusion-weighted magnetic resonance imaging (MRI) assigned more patients with EOC to the correct stage than computed tomography (CT) [5]. One-third of patients with serous EOC are diagnosed with stage IV disease [6], most commonly because of malignant pleural effusions or parenchymal liver and lung metastases [7]. A chest CT is performed as clinically indicated [1]; however, it has 14% sensitivity and 25% specificity for pleural status determination, when using video-assisted thoracic surgery as a reference [8]. Therefore, more reliable imaging tools for the upper abdomen and pleural space would be valuable.

Ultrasonography performed by an expert examiner might be a useful strategy for staging and treatment planning [9,10,11]. Recently, it was shown that a detailed transabdominal and transvaginal ultrasound examination, performed by an expert examiner, could render a similar diagnostic performance to CT for assessing pelvic/abdominal tumor spread disease in women with EOC [11]. Lung ultrasonography is a recognized imaging modality that can be successfully used in many conditions including cancer [12,13]. In a general population of patients with pleural effusions, one could suspect malignant effusions based on ultrasonographic findings [14]. Thus, comprehensive ultrasound staging with lung ultrasonography could be considered as an alternative imaging modality to CT scanning of patients with EOC in specific settings.

The present study aimed to describe lung and intercostal upper abdomen ultrasonography for pre-treatment staging of patients with EOC, demonstrate clinical examples and implications of this imaging, test its accuracy, and perform a feasibility study for future research.

## 2. Materials and Methods

A prospective observational study was conducted in a single tertiary cancer center from January 2017 to March 2019. The Ethical Committee of the Medical Council in Gdańsk, Poland approved the study protocol (KB-29/18, 23 Oct 2016), and all women gave informed consent.

### 2.1. Patients

Consecutive patients suspected of having EOC based on a subjective ultrasound assessment and scheduled for treatment in our institution, additionally underwent lung and intercostal upper abdomen ultrasonography for pre-treatment imaging. Exclusion criteria were final histological findings other than EOC, surgery was not performed, and ultrasonography was not performed by an expert examiner.

### 2.2. Imaging Technique

All ultrasound imaging was performed by a gynecologist-oncologist experienced in ultrasonography. According to our institutional guidelines, ultrasonography was the first and most important imaging modality for assessing patients suspected of having EOC. Patient preparations, such as enema and fasting, were not necessary, and no contrast agent was used. Abdominal and/or chest CT was performed only in selected cases, e.g., those with ambiguous ultrasound findings.

#### 2.2.1. Lung and Intercostal upper Abdomen Ultrasonography

The patient was placed in an upright sitting position, with the thorax exposed to the examiner. A Philips HD15^®^ ultrasound instrument (Philips Healthcare, Best, The Netherlands) was used. A convex 2.4-5-MHz probe was placed on the back thoracic wall, lateral from the spine. Firstly, the kidney was identified. Than the probe was moved up, held vertically and/or horizontally, crossing intercostal spaces. The liver was identified on the right side, the spleen on the left side. Next, the diaphragm was identified. Later, the probe was moved further up to the scapular bone, and pleural spaces and lungs were identified (air-filled lungs did not allow penetration of ultrasound and imaging of structures deeper than the pleural line, while pleural effusions and lung consolidation “allowed” more detailed examination of the area). After the first scanning, with the probe moved in vertical direction on the back thoracic wall, in the next steps the probe was applied to the intercostal spaces (probe parallel to ribs) in each lines of the body: vertebral, scapular, posterior axillary, median axillary, anterior axillary, midclavicular and sternal. The head was moved along the intercostal spaces and stopped in individual lines of the body, and then it was frequently stopped, angled, and tilted when spaces were crossed by body lines. We aimed to perform systematic scans of the diaphragm, pleural surfaces, lower parts of the pleural space, lungs, and upper abdomen, including the liver on the right side and the spleen on the left side. A diagram of the lung and intercostal upper abdomen ultrasound examination is presented in Figure 1.

Pathological ultrasound findings that could be detected in a supradiaphragmatic or subdiaphragmatic area were described as follows: pleural effusions (increased fluid in the pleural space), ascites (fluid between the diaphragm and liver/spleen); nodules (hyperechoic papillary projections) on pleural and abdominal surfaces of the diaphragm, bulky tumors (hyperechoic or mixed echogenic larger lesions with distinct borders), and plaque infiltration (mixed echogenic pathological area without distinct borders). We also searched for the involvement of the spleen/liver surface and hilum; parenchymal lesions in the liver, spleen, or lung; and enlarged cardiophrenic lymph nodes. No specific measurements or vascularization assessment was performed, but in a few cases, we used color flow Doppler imaging to differentiate small pleural effusions from pleural thickening. Videos and still images were recorded and discussed with experienced ultrasound operators (AB, NB, and JK).

#### 2.2.2. Transabdominal and Transvaginal Ultrasonography

Standard abdominal/pelvis ultrasound scanning was performed with a convex 2.4-5-MHz/transvaginal 5-9-MHz probe with the patient in the supine and gynecological positions, respectively, according to methodology and terminology described elsewhere [9,15]. Liver and spleen parenchyma, surface, hepatic and splenic hilum, upper abdomen, diaphragm, and peritoneum were scanned with both transabdominal and intercostal upper abdomen ultrasonography.

All ultrasound data were collected and managed using Research Electronic Data Capture (REDCap^®^, Vanderbilt University, Nashville, TN, USA) [16] before surgery and not changed thereafter. The ultrasound examiner was unaware of the results of any additional imaging, if performed.

### 2.3. Clinical Data

On the basis of comprehensive ultrasound imaging, we presumed the disease stage (The International Federation of Gynecologists and Obstetricians [FIGO] 2014), required surgical procedures, and the surgical complexity if primary debulking surgery was attempted. The calculation of surgical complexity was based on a scoring system described elsewhere [2] and facilitated with a web-based calculator (available online: http://gin-onc-calculators.com/lp.php). In accordance with imaging and patients’ clinical data, we suggested management strategies: upfront surgery or diagnostic laparoscopy followed by neoadjuvant chemotherapy.

### 2.4. Feasibility Study

In order to perform a feasibility study for future research on the diagnostic performance and clinical usefulness of lung and intercostal upper abdomen ultrasonography, we followed guidelines described elsewhere [17]. A feasibility studies are used to estimate important parameters that are needed to design the main study, e.g., standard deviation of the outcome measure, which is needed in some cases to estimate sample size, willingness of participants to be randomized, willingness of clinicians to recruit participants, number of eligible patients, characteristics of the proposed outcome measure and in some cases feasibility studies might involve designing a suitable outcome measure, follow-up rates, adherence/compliance rates, availability of data needed or the usefulness and limitations of a particular database, time needed to collect and analyze data [17,18].

### 2.5. Statistical Analysis

Statistical analysis was performed for ultrasound performance of combined transabdominal and intercostal upper abdomen ultrasonography for the subdiaphragmatic region, with a surgical-pathological examination as the reference standard. Statistical analyses were performed using the statistical suite STATISTICA version 12.0 (StatSoft. Inc., Tulsa, OK, USA) and Excel (Microsoft Corp., Redmond, WA, USA). To measure the diagnostic performance of ultrasonography, sensitivity, specificity, positive and negative predictive values, overall accuracy, the receiver operating characteristic curve, and area under the curve (AUC) were calculated (Appendix B). The statistical significance level of *p *< 0.05 was used.

The sample size for future research on performance of lung ultrasonography was calculated with an uncorrected chi-square statistic. Different scenarios were calculated based on available clinical and literature data at different statistical threshold parameters.

## 3. Results

### 3.1. Patients

During the study, 87 patients were eligible for inclusion. Ten patients were excluded because of the following reasons: non-EOC (*n *= 6) and surgery not diagnostic for evaluating the upper abdomen (*n *= 4). Ultimately, 77 patients were included. Median patients’ age was 60 years (range, 33–82 years). Histological findings indicated high-grade serous carcinoma in 52 (68%) women, endometrioid carcinoma in 11 (14%), mucinous in 5 (6%), clear-cell in 3 (4%), mixed in 4 (5%), and non-differentiated in 2 (3%). FIGO disease stages were I (*n *= 6 [8%]), II (*n *= 4 [5%]), III (*n *= 54 [70%]), IVa (*n *= 4 [5%]), and IVb (*n *= 9 [12%]). After excluding patients with FIGO disease stages I and IIA (*n *= 6) and those who underwent diagnostic laparoscopy because of poor performance status or comorbidities (*n *= 6), complete (R0) and optimal (R < 1 cm) cytoreduction was achieved in 40 (61%) and 7 (11%) patients, respectively. Median time from the ultrasound examination to surgery was 5 days (range, 1–14). All patients expressed their willingness to undergo the additional lung and intercostal upper abdomen ultrasound examination, and there were no technical problems associated with this procedure.

### 3.2. Imaging

Supradiaphragmatic disease was detected with lung ultrasonography in 13 cases. Sonographic features included pleural effusion on the right (*n *= 12) and left (*n *= 8) sides, nodular lesions on the diaphragmatic pleura (*n *= 9) (Figure 2a–d), lesions on the abdominal diaphragmatic surface (Figure 3a–d), focal lesion in lung parenchyma (*n *= 1), and enlarged cardiophrenic lymph nodes (*n *= 1) (Figure 4a–d). In most patients, the strong reflection of ultrasound waves from air-filled lungs did not allow penetration of ultrasound and imaging of structures deeper than the pleural line. It was not possible to test the performance of ultrasonography to evaluate the supradiaphragmatic area because not all patients with positive ultrasound findings had this area surgically explored, and those who had a negative imaging finding had the pleural cavity surgically explored occasionally (by incidental opening during peritoneal stripping of the diaphragm).

The performance of combined transabdominal and intercostal upper abdomen ultrasonography for subdiaphragmatic disease is shown in Table 1, and the AUC was 0.701–1.00, except for the left diaphragm and peritoneum (0.575). Specificity was more than 88% for the following evaluated regions: right and left diaphragm and peritoneum, hepatic and splenic hilum, and parenchymal lesions. Sensitivity values were 90% for the splenic hilum, 60% or less for the diaphragm and hepatic hilum, and 100% for liver and spleen parenchymal lesions; however, the latter two were true positive in 2 and 1 patients, respectively.

### 3.3. Clinical Implementation

Clinical examples of the implementation of lung and intercostal upper abdomen ultrasonography in pretreatment imaging are presented in Appendix C, and examples of images/loops with the corresponding surgical or CT presentations are shown in Appendix A (Appendix A, Appendix A, Appendix A, Appendix A, Appendix A, Appendix A, Appendix A, Appendix A and Appendix A).

### 3.4. Feasibility Study

In order to obtain meaningful results, the sample size for future research to evaluate the accuracy of lung ultrasonography was estimated at 160–240 cases, under the following conditions: type I error probability of 0.05, test power of 80%, and possible positive detection in 20–30% of consecutive patients with EOC.

## 4. Discussion

We described the use of lung and intercostal upper abdomen ultrasonography as a part of pretreatment imaging staging of patients with EOC. Supradiaphragmatic disease can be detected with lung ultrasonography, with pleural effusions being the easiest sonographic sign, and the “contrast” that enables detection of other larger lesions. Accuracy of a combined transabdominal and intercostal upper abdomen ultrasound examination in the subdiaphragmatic region was fair to excellent, except for the left abdominal diaphragm. Lung and intercostal upper abdomen ultrasonography enabled the presumption of stage IV disease, planning additional surgical procedures if primary cytoreduction was attempted, or modifying the initial management (upfront surgery or neoadjuvant chemotherapy). The feasibility study pointed to important obstacles that need to be addressed when planning a study to test the diagnostic accuracy and clinical usefulness of lung ultrasonography.

A detailed transabdominal/transvaginal examination performed by an expert examiner may offer similar diagnostic performance to CT for assessing abdominal tumor spread in women with EOC [11]. Overall sensitivities of ultrasonography and CT were 70.3% and 60.1%, respectively, and specificities were 97.8% and 93.7%, respectively. Compared to CT, ultrasonography had a slightly better sensitivity for pelvic regions and the omentum and worse sensitivity for assessing the small bowel; however, it was identical for assessing the root of the mesentery, mesogastrium, hepatic hilum, liver and spleen parenchyma, and retroperitoneal space [11]. Of note, after the initial exclusion of patients with extra-abdominal disease in this study, there were still 6 patients with stage IVa disease and 3 with stage IVb disease (probably parenchymal liver metastases). Ultrasonography underestimated 7 of 9 patients with stage IV (CT did not miss any) [11]. Perhaps physicians relying on lung and intercostal upper abdomen ultrasonography would assign more patients with pleural effusions and liver lesion to stage IV disease.

In another study comparing the diagnostic accuracy of transabdominal/transvaginal ultrasonography, CT, and whole-body diffusion-weighted MRI in EOC staging, ultrasonography showed the best results in disease detection in the pelvis and omentum; all three methods showed comparable results in the detection of the bowel surface and liver involvement, whereas ultrasonography had the lowest accuracy in the assessment of the diaphragm [19]. The sensitivity, specificity, and AUC of ultrasonography were 30.8%, 98.9%, and 0.648, respectively, for the detection of diaphragm carcinomatosis [10]. We showed that with combined transabdominal and intercostal upper abdomen ultrasonography of the right abdominal diaphragm, the sensitivity and AUC could be improved to 62.0% and 0.754, respectively. Nevertheless, still the rates of false negative were relatively high in the right and left side of diaphragm region (24.7% and 19.5%, respectively). This could be attributable to miliary peritoneal carcinomatosis—a form of a low volume disease, fine nodules that do not grow out significantly of the peritoneal surface (see Appendix A), thus are difficult to be detected with ultrasonography. Also, performance of ultrasonography was worse on the left diaphragm as compared to the right side (AUC 0.575 vs 0.754). This could be attributable to the fact, that there is an air-filled stomach on the left side, which may cause reflections of ultrasound waves. Moreover, the area between parenchymal organ and diaphragm on both sides is important in terms of imaging performance. The proximity and large area between the liver and diaphragm “enables” detection of peritoneal nodules that (if present) “disrupt” this otherwise smooth border. The spleen is small as compared to the liver, so the area between the diaphragm and parenchymal organ is smaller on the left side than on the right side. These could explain differences between left and right diaphragm imaging with ultrasound. From the surgical point of view, failure to detect low volume disease on the diaphragm peritoneum pre-operatively is not a major problem, because diaphragm peritonectomy with miliary carcinomatosis is relatively simple and quick procedure.

Ultrasonography of the pleurae and lungs, including the diaphragm, has been extensively described [12,13]. Detection and estimation of the pleural effusion volume using ultrasonography is reproducible and independent of the operator [12]. In the presence of pleural effusions, an acoustic window allows physicians to visualize the diaphragm and pleural cavity, thus a more detailed examination may reveal additional lesions. In the absence of free fluid, the phenomenon of a strong reflection of ultrasound waves from air-filled lungs does not allow penetration of ultrasound and imaging of structures deeper than the pleural line. Lung ultrasonography is a good diagnostic strategy to use for a transthoracic biopsy in the diagnosis of subpleural masses that are suspicious of malignancy [13], or to perform safe ultrasound-guided thoracocentesis [12]. For patients with advanced EOC, in whom pleural effusions or metastatic disease is suspected, video-assisted thoracoscopic surgery is considered a safe and accurate method for diagnosing metastases in the pleural cavity [8,20], and it can be planned immediately after lung ultrasonography.

Interestingly, there is ongoing research on the ultrasound evaluation of mediastinal lymph nodes with the use of transthoracic [21] and endobronchial and transesophageal ultrasonography [22]. Many other conditions, such as pneumothorax, atelectasis, pneumonia, pulmonary embolism, and cardiogenic pulmonary edema, can be diagnosed and evaluated using ultrasonography of the chest [13]. Patients with advanced EOC may suffer all aforementioned conditions.

Lung ultrasonography can save time and money, is readily available, and has no associated complications, side effects, or radiation exposure [12]. The examination can be performed early at the time of counselling and indicate the need for additional pretreatment work-up or serve as a single imaging modality. In contrast, performance of CT/MRI requires patient preparation, contrast intake, scheduling, and transportation to the CT/MRI departments. The estimated prices of abdominal/pelvic and pleural ultrasonography were 20% and 12% of the price of abdominal/pelvic and chest CT or MRI, respectively, in our institution.

The limitations of this study include its relatively small sample size, ultrasonography was performed by a single examiner, and it was not possible to systematically test positive and negative findings from lung ultrasonography with the surgical-pathological reference.

As for the feasibility of future research in the field of lung ultrasonography, some obstacles need to be addressed. Most studies analyzing the diagnostic accuracy of the imaging modality consider the surgical-pathological examination as the reference index. In the case of positive lung ultrasound findings in the supradiaphragmatic area, video-assisted thoracoscopic surgery seems the most appropriate procedure to serve as the reference standard, but it would test only true and false positive ultrasound findings. However, it would be difficult and unethical to submit patients for thoracoscopy in cases of negative lung ultrasound findings. Thus, the study design should rather include chest CT or MRI as the reference test for this area. Nevertheless, one must be aware that this solution would also lead to bias, as these imaging techniques are not as accurate as the surgical-pathological examination. In our setting, two patients had pleural nodular lesions detected with lung ultrasonography that were not detected by CT. Another issue for future study is that transabdominal/transvaginal findings should be recorded first, and lung and intercostal upper abdomen ultrasound diagnoses should be added separately in the second round of the examination—this procedure would enable testing of whether lung and intercostal upper abdomen ultrasonography adds value to transabdominal/transvaginal ultrasound staging and provides a clinical advantage. Moreover, intercostal ultrasonography is an additional procedure performed in a different patient position, and it requires extra time; thus, some clinicians might not be willing to recruit patients for such procedure. To test the possible clinical advantage/impact of lung and intercostal upper abdomen ultrasonography, a larger prospective trial should be conducted, and the following data should be collected: proportions of patients undergoing primary debulking surgery versus neoadjuvant chemotherapy and data about complete cytoreduction, adjuvant chemotherapy, disease-free survival, and overall survival.

## 5. Conclusions

Preoperative lung and intercostal upper abdomen ultrasonography performed in patients with EOC can add valuable information for disease spread in supradiaphragmatic and subdiaphragmatic regions, enable presumption of the disease stage and surgical complexity if upfront surgery is planned, or guide initial management. Performance of lung and intercostal upper abdomen ultrasonography is relatively easy (although it needs specific training), cost-effective, and does not require any special preparation. Therefore, it could be considered as additional or an independent imaging modality, especially in low resource settings. The accuracy of the method should be tested on a larger population, with the possible reference standard being thoracoscopy or eventually abdominal/chest MRI. A future area of investigation in the field should focus on improvement of the lung ultrasound technique by multidisciplinary interactions and training that involves scanning of the whole pleural cavity, lungs, and mediastinum, and consideration of contrast enhancement.

## Figures and Tables

**Figure 1 diagnostics-10-00085-f001:**
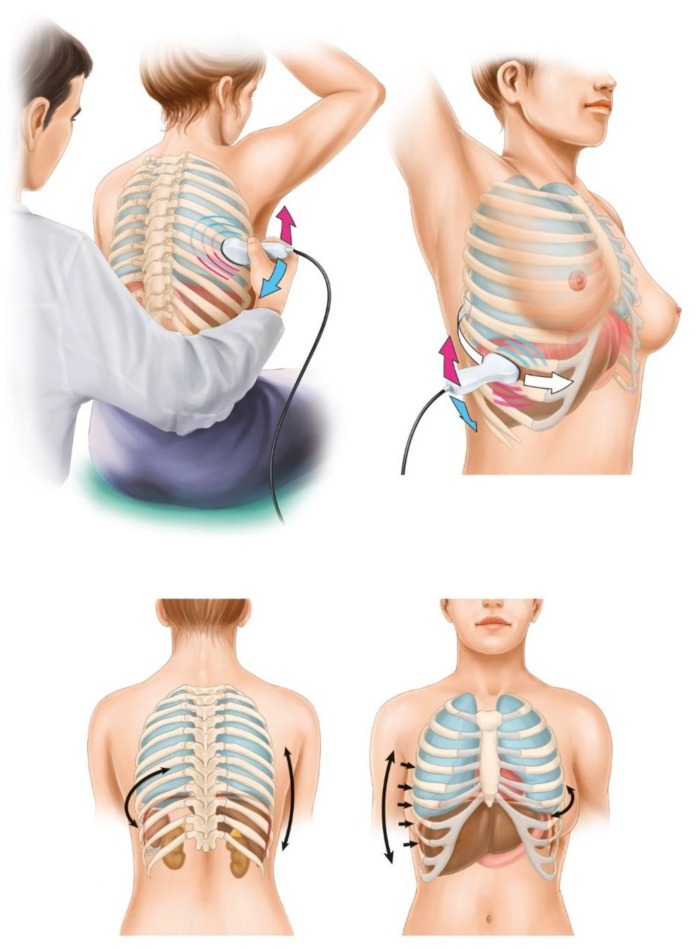
An illustration showing the technique of the lung and intercostal upper abdomen ultrasound examination (arrows indicate directions of probe application).

**Figure 2 diagnostics-10-00085-f002:**
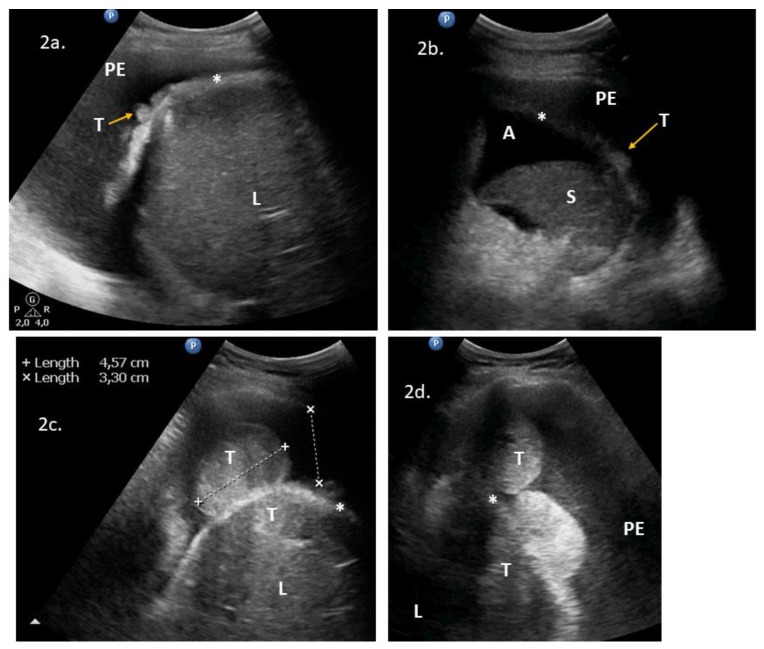
Lung ultrasonograms of lesions on the pleural diaphragmatic surface: (**2a**) nodules on the right diaphragmatic pleural surface and pleural effusions; (**2b**) pleural nodules and pleural effusions on the left side; and (**2c**) (**2d**) Bulky tumors on the right diaphragmatic pleural (and abdominal) surface and pleural effusions. Abbreviations: (*) diaphragm; (A) ascites; (L) liver; (PE) pleural effusions; (S) spleen; (T) tumor. Comment: The diaphragm is seen as a “bright line” and indicates the reflection between the air-filled lung and adjacent tissues. A normal diaphragm is 3–10 mm thick in the costal part and in the crus, respectively.

**Figure 3 diagnostics-10-00085-f003:**
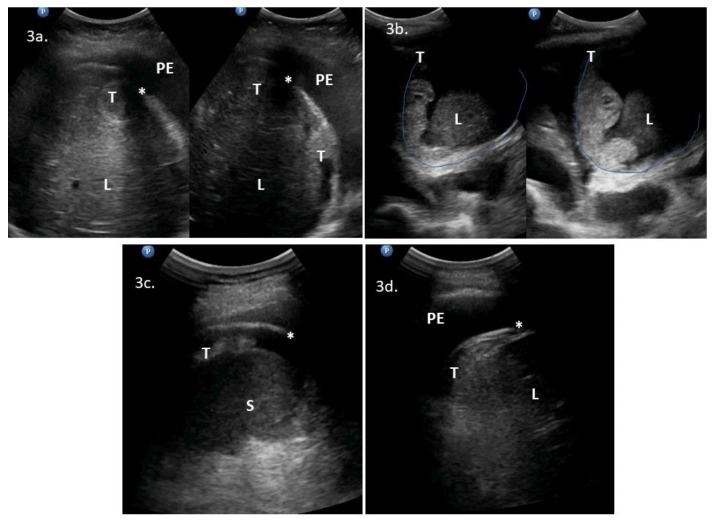
Intercostal upper abdomen ultrasonograms of lesions on the abdominal diaphragmatic surface: (**3a**) bulky tumors between the liver and diaphragm; (**3b**) solid-cystic tumors between the liver and diaphragm; (**3c**) tumor on the spleen surface; and (**3d**) plaque lesion on the right posterior abdominal surface of the diaphragm and pleural effusions. Abbreviations: (*) diaphragm; (L) liver; (PE) pleural effusions; (S) spleen; (T) tumor.

**Figure 4 diagnostics-10-00085-f004:**
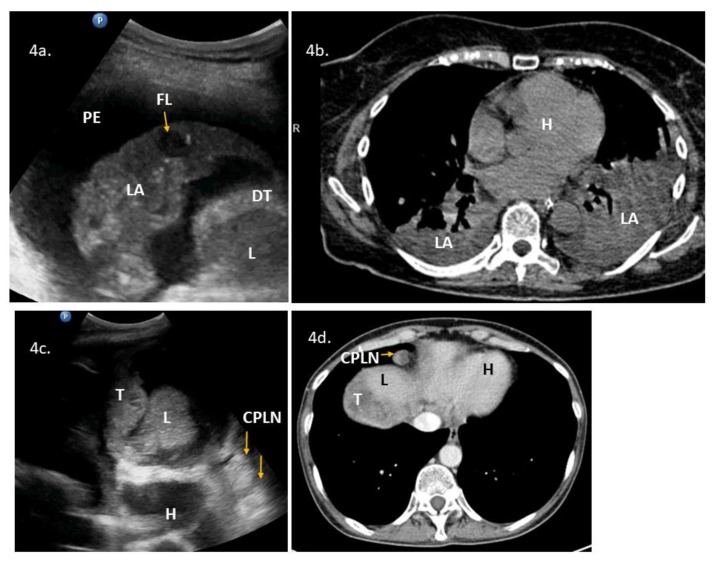
Lung ultrasonography and chest computed tomography (CT) of the lower parts of the pleural space and lungs: (**4a**) ultrasound presentation of lung consolidation, a sonographic air bronchogram with inflammation, and a metastatic parenchymal lung lesion (arrow, FL), pleural effusions, and diaphragm thickening; (**4b**) chest CT presentation of lung consolidation in the right and left lower lobes; and (**4c**) (**4d**) enlarged cardiophrenic lymph nodes (hyperechoic round lesions) on ultrasonography (3c) and chest CT (4d). Abbreviations: (CPLN) cardiophrenic lymph nodes; (DT) diaphragm thickening; (FL) focal lesion in the lung; (H) heart; (L) liver; (LA) lung with atelectasis; (T) tumor.

**Table 1 diagnostics-10-00085-t001:** Predictive parameters of preoperative combined transabdominal and intercostal upper abdomen ultrasonography for surgical-pathological findings in the subdiaphragmatic area.

	Sensitivity (95% CI)	Specificity (95% CI)	PPV (95% CI)	NPV (95% CI)	Overall Accuracy (95% CI)	TP (*n* (%))	FP (*n* (%))	FN (*n* (%))	TN (*n* (%))	AUC (95% CI)	*p*-Value
Liver, parenchymal lesions	100.0 (100.0;100.0)	98.7 (96.1;100.0)	66.7 (13.3;100.0)	100.0 (100.0;100.0)	98.7 (96.2;100.0)	2 (2.6)	1 (1.3)	0 (0.0)	74 (96.1)	0.993 (0.976;1.000)	<0.0001
Hepatic hilum	41.7 (13.8;69.6)	98.5 (95.5;100.0)	83.3 (53.5;100.0)	90.1 (83.2;97.1)	89.6 (82.8;96.4)	5 (6.5)	1 (1.3)	7 (9.1)	64 (83.1)	0.701 (0.509;0.892)	0.0403
Spleen, parenchymal lesions	100.0 (100.0;100.0)	100.0 (100.0;100.0)	100.0 (100.0;100.0)	100.0 (100.0;100.0)	100.0 (100.0;100.0)	1 (1.3)	0 (0.0)	0 (0.0)	76 (98.7)	1.00 (1.00;1.00)	<0.0001
Spleen hilum	90.0 (76.9;100.0)	94.7 (88.9;100.0)	85.7 (70.8;100.0)	96.4 (91.6;100.0)	93.5 (88.0;99.0)	18 (23.4)	3 (3.9)	2 (2.6)	54 (70.1)	0.924 (0.840;1.000)	<0.0001
Diaphragm, right side	62.0 (48.6;75.5)	88.9 (77.0;100.0)	91.2 (81.6;100.0)	55.8 (41.0;70.7)	71.4 (61.3;81.5)	31 (40.3)	3 (3.9)	19 (24.7)	24 (31.2)	0.754 (0.644;0.865)	<0.0001
Diaphragm, left side	16.7 (0.0;33.9)	98.3 (95.0;100.0)	75.0 (32.6;100.0)	79.5 (70.2;88.7)	79.2 (70.2;88.3)	3 (3.9)	1 (1.3)	15 (19.5)	58 (75.3)	0.575 (0.414;0.736)	0.3629

Abbreviations: AUC, area under the receiver operating characteristic curve; FN, false negative; FP, false positive; NPV, negative predictive value; PPV, positive predictive value; TN, true negative; TP true positive.

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
