# Peer review of "Lung and Intercostal Upper Abdomen Ultrasonography for Staging Patients with Ovarian Cancer: A Method Description and Feasibility Study"

_diagnostics, 2020, doi:10.3390/diagnostics10020085_

Round 1
Reviewer 1 Report
<General Comments>
The contents of this paper is interesting and useful in clinical site.
But it is hard for all gynecologists to examine lung and intercostal upper abdomen using this ultrasonography technique.
Please describe procedure and important points of matter to examine lung and intercostal upper abdomen.
<Specific Comments>
1) In Abstract: What did the authors compare the echo findings with? Is it surgical-pathological findings?
The authors should write it in Abstract.
2) In Discussion: Why was the accuracy rate low in the left abdominal diaphragm region? Please discuss.
The rates of false negative are relatively high in the right and left side of diaphragm region (24.7% and
19.5%, respectively). How do the authors overcome this? Please show the ways to solve the problems.
Author Response
Response to Reviewer 1:
Thank you very much for the review. We appreciate your time and work devoted to perform the review. We would like to thank you for your comments and suggestions.
A point-by-point response:
"Please describe procedure and important points of matter to examine lung and intercostal upper abdomen."
We added and edited the text – see lines 97 – 110.
"In Abstract: What did the authors compare the echo findings with? Is it surgical-pathological findings? The authors should write it in Abstract."
We added the information in the abstract – see lines 29 – 30.
"In Discussion: Why was the accuracy rate low in the left abdominal diaphragm region? Please discuss. The rates of false negative are relatively high in the right and left side of diaphragm region (24.7% and 19.5%, respectively). How do the authors overcome this? Please show the ways to solve the problems."
We added new text and discussed the issue – see lines 272 – 287.
The manuscript has been English language edited by professional editors at "Editage", a division of Cactus Communications. It was done before first submission to Diagnostics. See the attached file with the certificate.
Thank you.

Reviewer 2 Report
Dear authors, I read with interest your paper. The topic of the paper is interesting and timely for publication. The authors evaluated the clinical utility of lung and intercostal upper abdomen ultrasonography for staging patients with ovarian cancer and concluded that preoperative lung and intercostal upper abdomen ultrasonography could add valuable information for disease spread in supradiaphragmatic and subdiaphragmatic regions. Paper is well written. Study design is clearly stated. Technical aspects are explained precisely. Some limitations are clear. Mainly the small sample size of the study, and this is discussed.
Author Response
Response to Reviewer 2:
Thank you very much for the review. We appreciate your time and work devoted to perform the review. We are happy about your comments.
The manuscript has been English language edited by professional editors at "Editage", a division of Cactus Communications. It was done before first submission to Diagnostics. See the attached file with the certificate.
Thank you.
